# Gymnasium: A Standardized Interface for Reinforcement Learning Environments

**Mark Towers**[†]
University of Southampton &
Farama Foundation
mt5g17@soton.ac.uk

**Ariel Kwiatkowski**[†‡]
Meta AI, FAIR &
Farama Foundation
kwiat@meta.com

**Jordan Terry**[†]
Farama Foundation
jkterry@farama.org

**John U. Balis** [*]
Independent Researcher

**Gianluca De Cola** [*]
Farama Foundation

**Tristan Deleu** [*]
MILA, Université de Montréal

**Manuel Goulão** [*]
NeuralShift

**Andreas Kallinteris** [*]
Technical University of Crete

**Markus Krimmel** [*]
Farama Foundation

**Arjun KG** [*]
EarthBrain

**Rodrigo Perez-Vicente** [*]
Farama Foundation

**Andrea Pierré** [*]
UMass Lowell

**Sander Schulhoff** [*]
University of Maryland

**Jun Jet Tai** [*]
Coventry University

**Hannah Tan** [*]
Independent Researcher

**Omar G. Younis** [*]
MILA, Université de Montréal

## Abstract

Reinforcement Learning (RL) is a continuously growing field that has the potential to revolutionize many areas of artificial intelligence. However, despite its promise, RL research is often hindered by the lack of standardization in the environment and algorithmic implementations. This makes it difficult for researchers to compare and build upon each other's work, slowing progress in the field. Gymnasium is an open-source library that provides a standardized API for RL environments, aiming to tackle this issue, with over 18 million installations. Gymnasium's main feature is a set of abstractions that allow for wide interoperability between environments and training algorithms, making it easier for researchers to develop and test new environments and/or RL algorithms. In addition, Gymnasium provides a collection of built-in easy-to-use environments, tools for easily customizing environments, and tools to ensure the reproducibility and robustness of RL research. Through this unified framework, Gymnasium significantly streamlines the process of developing and testing RL algorithms, enabling researchers to focus on innovation and less on implementation details. By providing a standardized platform for RL research, Gymnasium helps to drive forward the field of reinforcement learning and unlock its full potential. Gymnasium is available online at https://github.com/Farama-Foundation/Gymnasium with documentation at https://gymnasium.farama.org/.

---

[†]Co-first authors
[‡]Contribution occurred prior to joining Meta
[*]Alphabetically ordered

39th Conference on Neural Information Processing Systems (NeurIPS 2025) Track on Datasets and Benchmarks.

# 1 Introduction

With the development of Deep Q-Networks (DQN, Mnih et al. [2013]), the field of Deep Reinforcement Learning (DRL) gained significant popularity as a promising paradigm for developing large-scale autonomous AI agents. Throughout the last decade, DRL-based approaches managed to achieve or exceed human performance in many popular games, such as Go [Silver et al., 2017], DoTA 2 [Berner et al., 2019], and Starcraft 2 [Vinyals et al., 2019].

During this time, OpenAI Gym [Brockman et al., 2016] emerged as the de facto standard open source API for DRL researchers. To implement RL environments, its simple structure and quality of life features made it possible to easily implement custom environments compatible with existing algorithm implementations. However, beginning in 2021, the project was no longer maintained by OpenAI staff, and no updates have been made since October 2022.

Gymnasium is the maintained successor to OpenAI Gym, which has become widely adopted, with over a million downloads in April 2025[*] and over 18 million downloads since its initial release in November 2023 to May 2025[*]. Since its development, over 500 GitHub issues have been raised[*], and 800 Pull Requests created to add features, fix bugs, or update documentation from over 40 unique contributors[*]. In this paper, we outline the main features of Gymnasium, the theoretical and practical considerations for its design, as well as our plans for future work. Collectively, we hope that Gymnasium removes barriers from DRL research and accelerates the development of safe, socially beneficial artificial intelligence.

In summary, Gymnasium provides the following novel contributions to the field of DRL:

- A maintained API for handling Reinforcement Learning Environment with a wide range of built-in environments (Figure 1), a collection of compatible external environments, and support from numerous training libraries.

- Supports a functional environment API for use in more theoretically aligned research, search algorithms, and use with vectorized hardware acceleration.

- Expanded support for vectorized environments with specialized wrappers, custom environment implementations, and API support for alternative vectorization methods and autoreset mode.

- Expansion of algebraic spaces to include text, graphs, disjoint unions, and variable-length sequences to enable multi-modal observation/action environments.

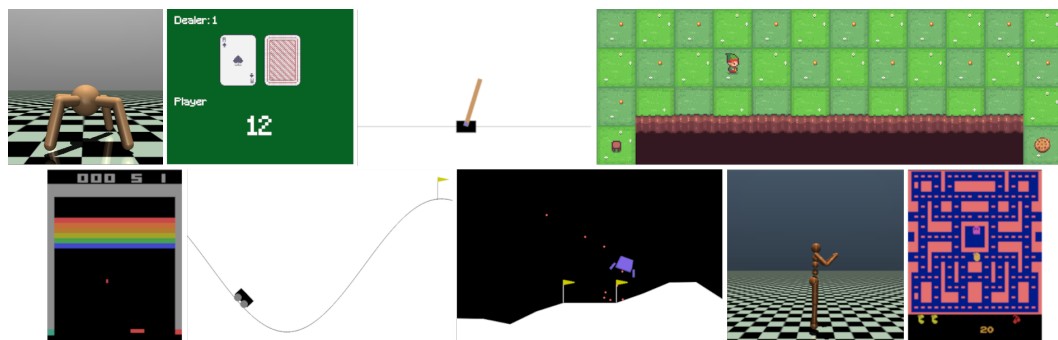

Figure 1: A subset of the available Reinforcement Learning Environments available through Gymnasium. From left to right, top to bottom: Ant-v5, Blackjack-v2, Cartpole-v1, FrozenLake-v1, ALE/Breakout-v5, MountainCar-v2, LunarLander-v3, Humanoid-v5, ALE/MsPacman-v5

---

[*]https://pypistats.org/packages/gymnasium
[*]https://pepy.tech/projects/gymnasium
[*]https://github.com/Farama-Foundation/Gymnasium/issues
[*]https://github.com/farama-Foundation/gymnasium/pulls

## 2 Related Work

Due to its popularity, Gymnasium enjoys a wide ecosystem of compatible libraries. In this section, we describe some of the most prominent RL libraries that can be used with Gymnasium, as well as alternative API libraries.

### 2.1 Training Libraries

**Stable Baselines3 (SB3)** [Raffin et al., 2021] is a popular library providing a collection of state-of-the-art RL algorithms implemented in PyTorch. It builds upon the functionality of OpenAI Baselines [Dhariwal et al., 2017], aiming to deliver reliable and scalable implementations of algorithms like PPO, DQN, and SAC. SB3 emphasizes usability and reproducibility, making it a natural fit for use with Gymnasium's environments.

**CleanRL** [Huang et al., 2022] is designed to provide clean, minimalistic, one-file implementations of RL algorithms. It focuses on simplicity and transparency, making it a helpful educational tool and easier for researchers to understand and experiment with RL techniques.

**Tianshou** [Weng et al., 2021] is a versatile library for RL research that supports various training paradigms, including off-policy, on-policy, and multi-agent settings. It offers a modular design that allows users to easily customize and extend the library's components.

**Ray Rllib** [Liang et al., 2018] is part of the Ray ecosystem and is known for its scalability and support for distributed RL training. Rllib provides a diverse range of algorithms and tools for both single-agent and multi-agent scenarios.

**Dopamine** [Castro et al., 2018] is a research framework developed by Google for experimenting with reinforcement learning algorithms. It is designed to provide a clean, minimalistic codebase that focuses on implementing and evaluating RL algorithms such as DQN and its variants in Tensorflow and JAX. Dopamine emphasizes reproducibility and simplicity, offering well-structured, modular components that make it easy for researchers to implement and test new algorithms.

### 2.2 Other Tooling utilising Gymnasium

**Minari** [Younis et al., 2024] defines a standardized format for offline RL datasets and provides a suite of tools for data management. These tools facilitate the collection, ingestion, processing, and distribution of datasets. Additionally, Minari integrates with a cloud-based repository that hosts a variety of benchmark datasets, enhancing accessibility and reproducibility in offline RL research.

**Atari Learning Environment** [Bellemare et al., 2013] is a collection of environments based on classic Atari games. It uses an emulator of the Atari 2600 console to ensure full fidelity and serves as a challenging and diverse testbed for RL algorithms.

**Metaworld** [Yu et al., 2019] is a benchmark for meta-RL and multi-task learning. It contains 50 robotic manipulation tasks compatible with the Gymnasium interface and allows for easy benchmarking of meta and multi-task algorithms.

**Gymnasium Robotics** [de Lazcano et al., 2023] is a collection of robotics environments, including maze path-finding and robot arm manipulation. It provides a multi-goal API compatible with Gymnasium, enabling support for algorithms like Hindsight Experience Replay [Andrychowicz et al., 2018].

### 2.3 Alternative API Libraries

Several alternative libraries offer different approaches to defining and interacting with RL environments.

**OpenAI Gym** [Brockman et al., 2016], the predecessor to Gymnasium, for which we built upon and extended while retaining its core principles. Still used in RL research, Gym is no longer being maintained and thus lacks the bug fixes, feature additions, and documentation compared to Gymnasium. As a result, few maintained RL libraries still support Gym anymore, though Gymnasium provides conversion wrappers to interface with Gym-based environments.

**dm_env** [Muldal et al., 2019] offers a lightweight API for RL environments, however, it is almost exclusively used within Google-Deepmind developed environments such as DM-Control [Tunyasuvunakool et al., 2020], DM-Lab2D [Beattie et al., 2020], and BSuite [Osband et al., 2020]. It emphasizes a clean, functional design and is used for evaluating algorithms in a controlled manner. While similar in some aspects to Gymnasium, dm_env focuses on providing a much more minimalistic API with a strong emphasis on performance and simplicity, thus lacking much of the additional tooling/capabilities that Gymnasium provides.

**PettingZoo** [Terry et al., 2021a] is designed for multi-agent RL environments, offering a suite of environments where multiple agents can interact simultaneously. It builds on concepts from Gymnasium but extends its capabilities to support complex multi-agent scenarios, making it an important tool for research in cooperative and competitive settings.

|  | **Gymnasium** | **OpenAI Gym** | **dm_env** |
|---|---|---|---|
| **Library Components** | Comprehensive with extensive wrappers, vectorization and spaces | Basic with several wrappers, vectorizations and spaces | Minimal with only lightweight utilities. |
| **Development** | Active | Unmaintained | Stable |
| **Community Adoption** | Widespread with native support across major RL frameworks. | Good though largely legacy | Limited with only ACME [Hoffman et al., 2020] and Dopamine [Castro et al., 2018] support |

Table 1: Comparison of RL Environment APIs

## 3 Library Structure

At its core, Gymnasium is a collection of interfaces tailored for RL environments. The simplicity and flexibility of the API enable it to be reused by researchers and developers across most environments and algorithms. In this section, we describe the main abstractions and interfaces included in Gymnasium with example code and tutorials in our documentation.

```python
import gymnasium as gym

env = gym.make("CartPole-v1", render_mode="human")
print(f"Observation Space={env.observation_space}")
print(f"Action Space={env.action_space}")

observation, info = env.reset(seed=42)
for _ in range(1000):
    action = env.action_space.sample()
    observation, reward, terminated, truncated, info = env.step(action)

    if terminated or truncated:
        print("New Episode Starting")
        observation, info = env.reset()
env.close()
```

**Environment** The central abstraction in Gymnasium is the Env representing an instantiation of an RL environment. An Env roughly corresponds to a Partially Observable Markov Decision Process (POMDP, Kaelbling et al. [1998]), with some notable differences discussed in Section 4.1. An Env is primarily defined by the attributes observation_space and action_space, and the methods reset and step. The observation_space and action_space define the expected observation and action structures for an environment (roughly corresponding to output and inputs expected). The reset method sets the environment to its initial state, beginning a new episode, and the step method executes a selected action in the environment, advancing it. For a detailed description of the Env class and example implementation tutorials, we refer readers to the documentation.

**Vector Environment**   A secondary, and arguably as important, abstraction is the `VectorEnv` that represents a collection of independently running `Env` with the results batched together. There are minimal structural differences with the same core attributes and methods to `Env` with the addition of `num_envs` for the number of sub-environments running in parallel. This enables relatively easy upgrading of training algorithms from `Env` to `VectorEnv` . The `VectorEnv` is crucial for modern RL research where algorithms use millions and sometimes billions of observations to learn from, as it enables hundreds or thousands of environments to be executed in parallel, maximising the steps per second taken.

**Space**   To define the expected structure of actions and observations for `reset` and `step` functions, spaces abstracts this with built-in implementations for matrix data, discrete categories, and more. They provide `dtype` and `shape` attributes that can be used for specifying a replay buffer's structure, a neural network's expected input observations and output actions, etc. They further implement a `sample` method to generate random examples, commonly used for random actions taken by agents.

**Registry**   To create and manage environments, Gymnasium has a registry that can be added to with a namespace, environment name, and a version number. For registered environments, they can be initialized with the `gymnasium.make` function, using arguments or wrappers also registered, allowing users to easily recreate an exact environment initialization. The standard Gymnasium convention is that any changes to the environment that modify its behavior result in incrementing the version number, ensuring reproducibility and reliability of historic RL research.

**Wrappers**   It's common for researchers to modify an environment's `reset` or `step` functions' inputs and outputs to change dynamics. Gymnasium provides `Wrapper` to easily achieve with 33 built-in wrappers that, for example, clip rewards, normalize observations, record MP4 videos, and more. See our list of wrappers for the complete set of built-in wrappers.

**Utility functions**   To further help researchers, Gymnasium provides a number of utility functions, such as an API checker to confirm that an environment is correctly implemented, for acting in an environment using a keyboard, and compatibility functions for operating with OpenAI Gym environments.

## 3.1   Implemented environments

Gymnasium includes a collection of well-tested environments fully compliant with the API (Figure 1) that can serve as references for new environment implementations or as simple testing grounds for algorithm development. For a benchmark of popular algorithms' performance against these environments, we refer readers to Huang et al. [2024], which provides training curves and benchmarks from several training libraries. Here, we describe these environments and their general uses.

**Classic control**   Originally inspired by textbooks or historic RL research, Cartpole, Acrobot, Mountain Car, and Pendulum are simple physics simulations that provide easy testing environments with continuous observation spaces and either discrete or continuous actions. These environments tend to serve well as quick and simple evaluations for new algorithm implementations due to their simplistic reward structures and small observation and action spaces. A notable exception is Mountain Car, which is often difficult to solve without a deliberate exploration mechanism due to its sparse rewards.

**Toy text**   Modelled on tabular MDPs (Markov Decision Process, Puterman [1990]), Blackjack, Taxi, Cliff Walking, and Frozen Lake are all discrete observation and action environments. As such, these environments can serve as testbeds for RL algorithms that do not rely on neural networks for state representation.

**Box2D**   For more complex physics simulations, Bipedal Walker, Car Racing, and Lunar Lander use the Box2D physics engine with collision detection and contact forces (both of which are neglected in the classic control environments). These environments use continuous or image-based observations and continuous or discrete actions, and represent a step up in complexity compared to classic control and toy text environments.

**MuJoCo**  Robotics-based problems with higher dimensionality and non-linear dynamics, the MuJoCo environments are eleven physics-based environments with continuous sensors (observations) and control (actions) using the MuJoCo simulator [Todorov et al., 2012]. These environments simulate more complex physical interactions, including multi-body dynamics and contact forces, offering a more realistic and challenging setting for RL agents to learn. Each environment involves controlling a simulated agent to achieve tasks such as locomotion or balancing, and is still commonly used as a suite of benchmark environments for continuous control algorithms.

**External environments**  In addition to the built-in environments listed above, due to Gymnasium's popularity, numerous projects incorporate our API, for which we maintain a list of external environments which are frequently updated. We categorise these into first-party environments, maintained by the Farama Foundation, and third-party environments. The list of first-party environments includes gridworlds [Chevalier-Boisvert et al., 2023], robotics [de Lazcano et al., 2023], web-based [Liu et al., 2018], arcade games [Bellemare et al., 2013, Kempka et al., 2016, Poliquin, 2025], meta-objectives [Yu et al., 2019, Felten et al., 2023], and automated driving [Leurent, 2018] environments. For third-party environments, they include automated driving [Book et al., 2023, Alegre, 2019, Bouteiller et al., 2025], medical [Choudhary et al., 2024], UAV control [Gong et al., 2025, Groot et al., 2024, de Moura Souza and Toledo, 2024, Tai et al., 2023], robotics [Malagón et al., 2024], and many more environments. Additionally, work has built upon these base environments for more specialist research, for example further environments using Atari [Delfosse et al., 2024, Shao et al., 2022, Dalton et al., 2020, Young and Tian, 2019, Terry et al., 2021b].

# 4   Novel features

Since its fork of OpenAI Gym, over 800 Pull Requests by over 40 unique contributors have been opened in Gymnasium to add features, fix bugs, and improve documentation. Beyond the quality of life and ease of use changes implemented, in this Section, we highlight four core changes to the API.

## 4.1   Functional Environment API

The `Env` class is the central abstraction for Gymnasium with an object-oriented design. More recently, the `FuncEnv` was added as a secondary abstraction for implementing environments with a more functional approach. Inspired by RL environment theoretical formalism, POMDP [Kaelbling et al., 1998], the `FuncEnv` defines functions that correspond directly to POMDP components (unlike `Env`):

- `initial(rng) -> state`: Generates the initial state of the environment, mirroring $\mu \in \Delta\mathcal{S}$ that samples the initial state distribution where $\mathcal{S}$ is the set of possible states.

- `transition(state, action, rng) -> state`: Computes the next state of the environment based on an action, matching $T\colon \mathcal{S} \times \mathcal{A} \to \Delta\mathcal{S}$ that transitions a state to the next state given an action, where $\mathcal{S}$ and $\mathcal{A}$ is the set of possible states and actions, respectively.

- `observation(state) -> obs`: Returns the observation for a given state in the environment, following $O\colon S \to \Delta\Omega$ that maps states ($S$) to observations ($\Omega$).

- `reward(state, action, next_state) -> float`: Computes the reward for transitioning from one state to another given an action, mirroring $R\colon \mathcal{S} \times \mathcal{A} \times \mathcal{S} \to \mathbb{R}$ where $S$ and $A$ are the set of possible states and actions respectively, and $\mathbb{R}$ is set of real numbers for the reward.

- `terminal(state) -> bool`: Determines if a state is terminal, indicating the end of an episode. This does not correspond to any element of the classical POMDP description, but is necessary for practical reasons. We expand on this in the following section

This design has two main advantages: first, that it's more closely aligned to the theoretical POMDP formalism. This is important as with the standard `Env` formulation, theoretical and search-based RL research is more difficult to accomplish, as understanding how an environment's state evolves over time, the impact of stochasticity in transitions, and more is significantly more convoluted due to `Env`'s object-oriented implementation. Instead, with `FuncEnv`, users have direct access to the environment's state, for which multiple transitions can be taken to collect a distribution of possible next states, often necessary in theoretical and search-based research. Secondly, `FuncEnv` allows easier

hardware acceleration of implemented environments using libraries like JAX [Bradbury et al., 2018]. Scaling `Envs`, discussed further in Section 4.3, is difficult as it normally requires unique instances of each environment that are iteratively computed. In comparison, the functional paradigm of `FuncEnv` enables greater scalability as JAX [Bradbury et al., 2018] and other libraries can efficiently vectorize stateless functions across CPUs and given GPUs with minimal changes. Brax [Freeman et al., 2021], Pgx [Koyamada et al., 2023], and Jumanji [Bonnet et al., 2024] are all example projects that have been built around JAX to parallelize their environment functions. To support this, Gymnasium provides the `FunctionalJaxVectorEnv` class to vectorize any `FuncEnv` written in JAX. As such, the `FuncEnv` provides an alternative, though highly effective, environment API for environment implementations compared to `Env`. However, to unlock all these advantages for a `Env` requires that it be reimplemented as a `FuncEnv` due to the API differences.

## 4.2 Termination and Truncation

In RL theory, it is common to assume that actions can be executed in an environment indefinitely. In practice, this tends to be unrealistic, as researchers only have a finite time to perform their experiments. To account for that, Gymnasium introduces the notions of episodic termination and truncation, visualised in Figure 2. In OpenAI Gym, these two concepts were not clearly separated due to the `Env.step` implementation, so it is worth expanding on their meaning, the distinction between them, and how Gymnasium helps clarify the issue.

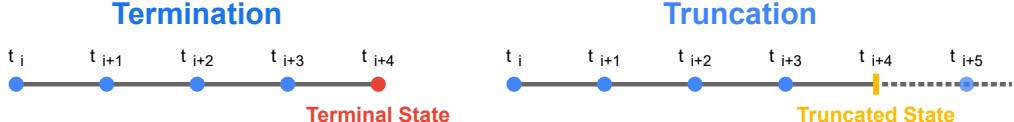

Figure 2: Example of termination and truncation

**Termination** is a signal for a state to indicate if an agent cannot reach any further states from it. Typically, this represents either the success or failure of the task that an RL agent is trying to achieve, but generally, it's any state-dependent reason to end an episode. It is possible for an environment not to have any terminal states, as is common in theoretical RL problems where environments enter an absorbing state with 0 reward. However, practically most Gymnasium environments have termination signals.

**Truncation** similarly indicates the end of an episode, but for a reason unrelated to the environment's current state. This is normally based on the number of steps taken, but could be used if the environment crashes or times out for reasons unrelated to the agent's behaviour. It is roughly the equivalent of saying "The episode could keep going, but we ran out of time, so we decided to stop it here" as shown in Figure 2. In Gymnasium, we provide the `TimeLimit` wrapper, through which developers and users can define the number of steps after which the environment will give a truncation signal. Practically, like Termination, most Gymnasium environments have a maximum action taken value before truncation to prevent an agent from getting stuck and never terminating.

While the difference between termination and truncation may seem minor, it has nontrivial implications for algorithm implementations. For algorithms that compute the expected future rewards, through estimating state-value, $V(s)$ like REINFORCE [Sutton et al., 1999] and Proximal Policy Optimization [Schulman et al., 2017] and the Q-vaule, $Q(s, a)$ like Deep Q-Networks [Mnih et al., 2015] and Soft Actor-Critic [Haarnoja et al., 2018]. [*]

$$V(s_t) = r_t + \gamma \cdot \neg \text{ terminated}_t \cdot V(s_{t+1}) \tag{1}$$

$$Q(s_t, a_t) = r_t + \gamma \cdot \neg \text{ terminated}_t \cdot \max_{a^* \in A} Q(s_{t+1}, a^*) \tag{2}$$

Computing the expected future rewards, if an episode terminates, then no further rewards can be obtained after the terminal state. In contrast, if an episode truncates, then the agent has encountered

---

[*]$s_t, a_t, r_t$ is the state, action taken, and reward received on timestep $t$, and $s_{t+1}$ the state for timestep $t + 1$. $\gamma$ is the discount factor to encourage collecting temporally sooner rewards.

an arbitrary cutoff, and if not, would have kept acting, accumulating rewards. As a result, in the value estimation algorithm (Eqs. (1) and (2)), whether an episode terminates or truncates is crucial to the predicted expected future rewards and for agent policies.

However, for OpenAI Gym users, differentiating between whether an environment has terminated or been truncated was a single boolean signal, with the developer needing to check a `step`'s information if a truncation occurred. As a result, few users and even well-developed training libraries correctly differentiated between the two.[*] To address this issue, Gymnasium modified `Env.step`'s type definition to return two boolean signals, one for termination and one for truncation. This forces users to acknowledge when termination vs truncation occurred, with the plan that this would improve training libraries that utilize Gymnasium.

## 4.3 Vectorization

Gymnasium puts an emphasis on treating vectorized environments as first-class citizens, on par with individual environments. This is because vectorization is a common optimization technique in RL research that allows running numerous independent environments in parallel, enabling significant performance gains without major changes to an algorithm's implementation.

By default, Gymnasium supports two vectorization modes, as well as a custom mode that can be defined for each environment separately. `SyncVectorEnv` vectorizes arbitrary environments very simply through iteratively computing each sub-environment's result and batching it together. `AsyncVectorEnv` running each sub-environment in its own subprocess, multithreading them at the cost of greater memory requirements. As a result, the relative performance of `SyncVectorEnv` and `AsyncVectorEnv` vectorization depends on many factors, notably the complexity of the environment itself and the hardware used. For example, Figure 3 shows the steps per second of the Box2D Lunar Lander environment, vectorized for high-end and low-end hardware.

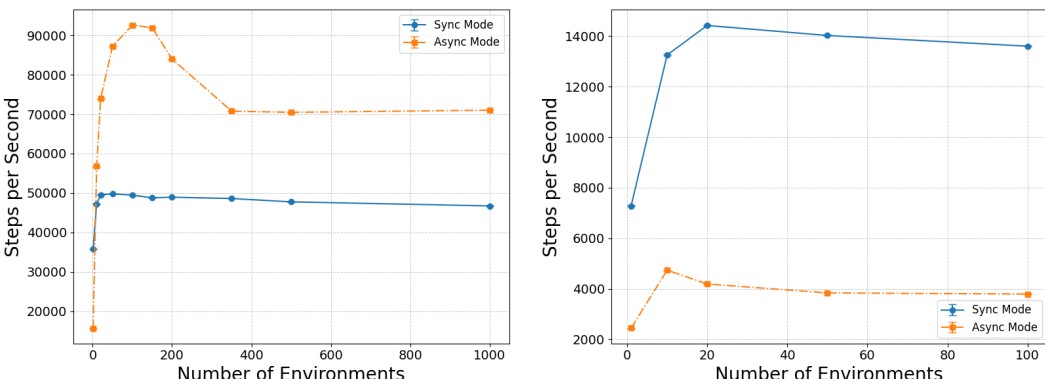

(a) Individual environment steps per second, executed on a MacBook Pro. Higher is better.

(b) Individual environment steps per second, executed on Google Colab. Higher is better.

Figure 3: Performance comparison of different vectorization modes on the Lunar Lander environment using 10,000 total environment steps. On a system with high RAM (M3 MacBook Pro with 128 GB of RAM), `Async` vectorization provides performance benefits by running environments in parallel. Conversely, on weaker hardware (Google Colab), the subprocess overhead remains significant and significantly degrades `Async` performance.

In addition, Gymnasium added the ability to implement and create custom vector environments that can implement their own custom vectorization method. An example is classic control CartPole, where, because of its simple dynamics, sub-environments can be all computed at the same time using tensor operations in parallel. This increases the steps per second by an order of magnitude faster computation compared to both `SyncVectorEnv` or `AsyncVectorEnv` and without the memory overhead of separate sub-environment instants or subprocesses, etc. However, this requires the environment to be custom-written and isn't normally compatible with more complex environments.

---

[*]`https://farama.org/Gymnasium-Terminated-Truncated-Step-API`

Though this opens up the opportunity to researchers to develop their own custom vector environments like Kazemkhani et al. [2025], Weng et al. [2022], Suarez [2024].

A further critical implementation detail for vector environments is that sub-environments will terminate (or truncate) after a different number of steps. Therefore, to maximise the throughput, it's common for vector environments to automatically reset sub-environments whose episodes end, referred to as Autoreset. As Figure 4 demonstrates, there are three distinct methods for auto-resetting, for which OpenAI Gym only supports same-step autoreset. The difference in their implementation is when the sub-environment is reset, for next-step mode this occurs on the timestep after the last episode ended such that every timestep a sub-environment is only either reset or step. This is the Gymnasium default and fastest option as a sub-environment is only completing one option at a time however for reset timesteps then there are "dead" actions that aren't passed to an environment. Second, in contrast, for the same-step mode when a sub-environment's episode ends then its reset within the same timestep and the episode's last observation is returned within the step's info data. This was the OpenAI Gym default and requires that users correctly use the observation in info to value estimation and prevents rendering of an episode's final observation. Thirdly, the disabled mode requires an explicit reset call using a mask to specify which sub-environments are reset. This provides the most control to users about which sub-environments are reset and when. We've added support for all three reset modes into our build-in vectorizers and vector wrappers where compatible.

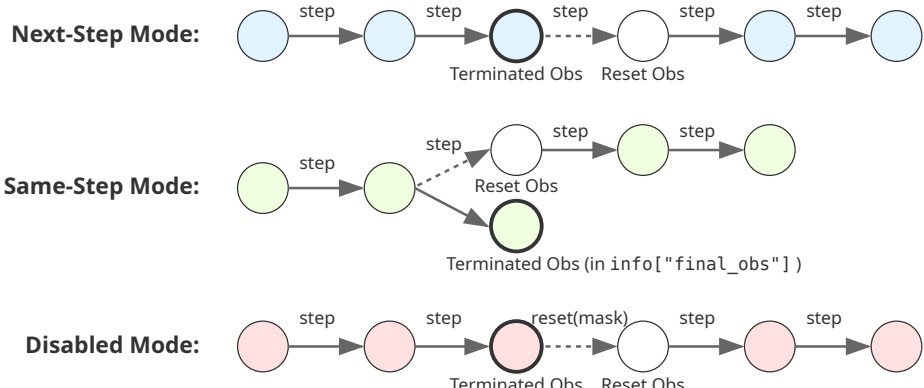

Figure 4: Methods of auto-resetting vectorized sub-environments. The white circles are the reset observation, and the circles with bold outlines are an episode's final observation. The solid arrow is for sub-environment steps, and the dashed arrow for sub-environment resets.

## 4.4 Algebraic spaces

In Gymnasium, observation and action spaces can be broadly split into two categories: fundamental and composite. As the name suggests, composite spaces comprise one or more subspaces, whereas fundamental spaces cannot be divided this way. Gymnasium supports the following built-in spaces, with the Text, Sequence, Graph, and OneOf being newly added and are unique to Gymnasium:

These spaces provide researchers and users the capability to design and implement multi-modal reinforcement learning where an agent may take input signals (observations) from a network of connected devices using the Graph space, or a variable length of visible obstacles that must be avoided using the Sequence space.

## 5  Summary

This paper introduces Gymnasium, an open-source library offering a standardized API for RL environments and a suite of benchmark environments ranging from finite MDPs to robotic simulations. As a result, it has wide-scale adoption and compatibility with training libraries and external environments. Building upon OpenAI Gym, beyond quality of life and ease of use changes, we have introduced new features to accelerate RL research, such as an emphasis on vectorized environments, new spaces

**Fundamental Spaces**

- **Box** : for multidimensional arrays of discrete or continuous values
- **Discrete** : for single integers
- **MultiBinary** : for multidimensional arrays of binary values
- **MultiDiscrete** : for sequences of discrete or categorical options
- **Text** : for strings

**Composite Spaces**

- **Dict** : for (Python) dictionaries of spaces
- **Tuple** : for tuples of spaces
- **Sequence** : for variable-length sequences of spaces
- **Graph** : for graphs with nodes and edges specified by spaces
- **OneOf** : for disjoint unions of spaces

Figure 5: List of spaces split into the Fundamental and Composite

for multi-modal observations or actions, and an explicit interface for functional environments that can be hardware-accelerated. These tools have the goal of accelerating the advancements of safe and beneficial AI research.

However, there are two primary limitations to Gymnasium. The first is that, as it is solely an RL environment API and thus requires users to train an agent to use a separate training library, like those listed in Section 2.1. Secondly, Gymnasium is implemented in Python due to its popularity in machine learning research; however, as a result, environments can be slow to run compared to custom C, C++, or Cython-based environments [Suarez, 2024].

## Acknowledgments and Disclosure of Funding

Thank you to the numerous contributors and users of Gymnasium who have helped develop the project over the years, in particular, the original OpenAI team that created Gym.

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
