# OpenReview forum: "Gymnasium: A Standard Interface for Reinforcement Learning Environments"
_NeurIPS.cc/2025/Datasets_and_Benchmarks_Track — NeurIPS 2025 Datasets and Benchmarks Track spotlight_

### Official Review · Reviewer_ngtH · 2025-06-03

**Rating:** 6
**Confidence:** 5

**Summary:**

The paper introduces Gymnasium, a standardized API designed to improve interoperability and reproducibility in reinforcement learning (RL) research. Gymnasium, an update and extension of the OpenAI Gym, addresses issues such as inconsistent environment implementations and the need for easy experimentation and robust testing. Its main contributions include a unified API compatible with multiple RL libraries, support for both individual and vectorized environments, and novel features such as a functional API closely aligned with Partially Observable Markov Decision Processes (POMDPs), a structured approach to termination and truncation, and expanded support for algebraic spaces. Gymnasium also provides a suite of customizable benchmark environments that range from simple tabular MDPs to complex physical simulations using MuJoCo, helping to accelerate RL algorithm development and testing.

**Additional Feedback:**

- It would be better if the authors could include more RL benchmarks as references
- It would be better if the authors could compare the Gymnasium with other RL environments in a tabular format
- On page 1, the footnote should be hidden if the authors choose to submit this paper anonymously.

**Dataset Code Accessibility:**

Yes

**Dataset Code Comments:**

Gymnasium provided sufficient detail and API for DRL research

**Ethical Considerations:**

No, there are no or only very minor ethics concerns

**Final Justification:**

I keep the same rating and recommend a strong acceptance of this paper. This is a well-known work and definitely should be accepted by NeurIPS.

**Limitations Weaknesses:**

The quality of the figures could be significantly improved. Please try to export pdf with matplotlib instead of taking screenshots or using png/jpeg figures by convention.

**Strengths Contributions:**

- By providing a consistent API, Gymnasium streamlines RL research, making it easier for researchers to compare and build upon previous work. The functional API enhances hardware-accelerated environments using libraries like JAX, which is beneficial for large-scale or computationally intensive applications.
- The built-in vectorization features (Sync and Async) support efficient parallelization of environments, which can significantly improve the performance of RL training processes.
- Gymnasium includes environments from basic MDPs to complex MuJoCo simulations, catering to various RL research needs and supporting a broad range of algorithms and approaches.

Overall, this paper definitely should be accepted and highlighted in NeurIPS.

---

> ### Author Rebuttal · Authors · 2025-07-31
>
> Thanks for the review and understanding the strong contributions of the project.
>
> We’ll upgrade the figures to use PDF or SVG
>
> For the additional feedback
> 1. This was a deliberate decision not to include the performance of different training algorithms for the included environments, as the library isn’t a training library, therefore would need to rely on another paper's / project's results. We do refer users to OpenRLBenchmark, which provides a more comprehensive evaluation for a range of Gymnasium environments and training libraries.
> 2. Yes, we can look to make the API comparisons into a tabular format
> 3. Apologies about that, we missed that
>
> We hope these answer your questions.

---

> > ### Comment · Reviewer_ngtH · 2025-08-01
> >
> > Thank you for answering my questions, and thanks for your contributions to the RL community! I have no further comments and recommend a strong acceptance of this paper.

---

### Official Review · Reviewer_KqRy · 2025-06-14

**Rating:** 4
**Confidence:** 5

**Summary:**

This paper describes the Gymnasium framework (in particular: the most important changes since OpenAI stopped supporting the previous iteration called Gym, and Farama Foundation took over).

**Additional Feedback:**

For the functional API described in 4.1: is my assumption correct that this is not "automatic"? As in, every single existing environment will have to be manually re-written for the functional API, if we want it to be available via the functional API? Based on my knowledge of how the implementations work, I imagine it'd be very difficult not to have it be like this, but would be nice to explicitly mention it.

**Dataset Code Accessibility:**

Yes

**Dataset Code Comments:**

Code is available (and already has been downloaded millions of times and is used by essentially the entire RL research community).

**Ethical Considerations:**

No, there are no or only very minor ethics concerns

**Final Justification:**

Copied from my public comment to the authors:

>While you acknowledge and promise to address the key parts I pointed out, at this point they're just promises to fix clarity issues. That is not a solid enough basis for me to increase my score.
>
>Even with my current score, I will not argue against acceptance, as I'm still on the accept side of the threshold. That I'm only there at a borderline level probably won't matter in practice, as all the other reviewers are already at higher levels. However, as the only changes that I requested are textual ones, I suppose if you have enough time you could show a bit more thoroughly exactly what your updated text will look like (by writing the planned additions directly in OpenReview). That could provide enough basis for me to update to a score higher than Borderline Accept.

As the authors did not provide a further response, I did not adjust my score.

**Limitations Weaknesses:**

1. A bit sloppy writing in some places. For example: final sentence of page 6 seems to be unfinished (it's already a very long sentence too, but not actually finished). Line 316 has a sentence that doesn't end in a period.
2. Figure 4 is... mostly clear, but there's actually not a real explanation of it. The paper only mentions that there are these three different ways of resetting, but doesn't explain them. It would also be nice to explain how to read the figure (e.g., what's the significance of white circles? they seem to have a special meaning?). This is one of these key new things in Gymnasium that was not in original Gym, so it's sad not to see it properly explained.
3. Point 7 in the Checklist claims that Figure 3 has error bars. This is not true. At least, I can't see them. Maybe they're just very small. Regardless, I'm also missing an explanation of how they are calculated (if they are at all), and other details surrounding this experiment (e.g., exactly how was each datapoint measured? Over how much time / how many steps?)
4. There is a curious disconnect between the very explicitly expressed hope that Gymnasium "accelerates the development of safe, socially beneficial artificial intelligence" in the Introduction, and the claim that "We don't believe there is a societal impact of the work performed." in point 10 of the Checklist.


I expect that many/all of my comments should be relatively easy to address, and would be happy to raise my score if they are (and/or if it's pointed out how I'm wrong). I suppose that weaknesses (2) and (3) are the most important ones in this respect. It is crucial that an academic paper is easily understandable in its entirety, including all figures. And for experiments, it is always crucial that all the details around them are fully listed, and of course some degree of analysis of statistics is key (there's a reason why it's an entry in the checklist).

**Strengths Contributions:**

- A very important framework: API is used and/or built on by practically the entire RL research community
- While there already of course is the original gym paper from 2016, there have been some important additions to Gym (now Gymnasium) which as far as I know have indeed not yet been properly detailed in any academic publications. This paper addresses that gap.

---

> ### Author Rebuttal · Authors · 2025-07-31
>
> Thanks for the review and recognising that basically the whole RL community has shifted across to Gymnasium and the value of this paper.
>
> Answering Limitations / Weaknesses
> 1. We’ll fix the sentence on page 6 and line 316
> 2. We’ll improve the description of Figure 4 and how to understand it.
> 3. Figure 3 does have error bars, but they are very small; we’ll add a description for how the error bars were created.
> 4. Thanks for recognising the disconnect between our answers. We’ll update point 10 of the checklist to reiterate of hope for Gymnasium to be used to accelerate the development of safe, socially beneficial artificial intelligence.
>
> For the additional feedback
> * Yes, you're correct that environments will have to be rewritten for the functional API and will add a point to make this clear to readers.
>
> We hope these answer your questions.

---

> > ### Comment · Reviewer_KqRy · 2025-08-02
> >
> > Thanks for your responses.
> >
> > While you acknowledge and promise to address the key parts I pointed out, at this point they're just promises to fix clarity issues. That is not a solid enough basis for me to increase my score.
> >
> > Even with my current score, I will not argue *against* acceptance, as I'm still on the accept side of the threshold. That I'm only there at a borderline level probably won't matter in practice, as all the other reviewers are already at higher levels. However, as the only changes that I requested are textual ones, I suppose if you have enough time you could show a bit more thoroughly exactly what your updated text will look like (by writing the planned additions directly in OpenReview). That could provide enough basis for me to update to a score higher than Borderline Accept.

---

> > > ### Comment · Area_Chair_wWuM · 2025-08-08
> > >
> > > @KqRy -- you need to still acknowledge that you read the rebuttal.

---

### Official Review · Reviewer_RBDF · 2025-06-23

**Rating:** 6
**Confidence:** 5

**Summary:**

This paper introduces the famous gymnasium library. Built upon the deprecated gym library, gymnasium normalizes the usage of different RL environments, and provides multiple wrappers (allowing for parallelization, recording, ...etc) and already incorporates many built in and third party envs.
The paper clearly explains the implementation choices and their advantages and drawbacks.

**Additional Feedback:**

I have been using gymnasium for many years, and am an independent developer of environments built on top of it. Thank you for your open source work.

I came across [this post](https://psc-g.github.io/posts/research/rl/atari_defense/) lately, it could be interesting to discuss that even subfields of gymnasium have emerged (as you have remaining space).
**In Atari**:
* [Continuous ALE](https://arxiv.org/pdf/2410.23810v1)
* [Multiplayer ALE](https://arxiv.org/pdf/2009.09341)
* [Mask Atari](https://arxiv.org/pdf/2203.16777)
* [Object-centric Atari](https://arxiv.org/pdf/2306.08649)
* [HackAtari](https://arxiv.org/pdf/2406.03997)

**In control**:
* The Gymnasium robotiques suites.
* [PandaArmControl](https://arxiv.org/pdf/2203.16777)
* [ControlGym](https://arxiv.org/pdf/2311.18736)
* [RoboSuite](https://arxiv.org/pdf/2009.12293)
... etc.

I guess this can also be done for board games and other subfields.

Questions:
* Will the GPU emulated Atari be integrated in gymnasium at some point ?
* It seems that ALE is split out. At least the ALE documentation is hosted on another website. Will ALE remain within gymnasium?

**Dataset Code Accessibility:**

Yes

**Dataset Code Comments:**

No challenge, the links to the open source implementation and documentation are provided.

**Ethical Comments:**

NA.

**Ethical Considerations:**

No, there are no or only very minor ethics concerns

**Final Justification:**

This paper introduces the most used RL environment, the authors have addressed the reviewers concerned. This papers deserves at least a spotlight, if not an oral.

**Limitations Weaknesses:**

Maybe provide in the paper as well a link/explanation on how to add a third party environment.
Please provide vectorial images for figures 2, 4 and 5.

Minor typos and suggestion:
You could use a 3rd Atari frame in the bottom row (maybe of a 3D game e.g. BattleZone or Robotank), or you can crop BlackJack to avoid losing space and group them.
Then you can provide a better caption:
**A subset of the available reinforcement learning environments available through Gymnasium.** Gymnasium maintains classic control, box2D, toy text (e.g. *Blackjack* and *CliffWalking*), mujoco-based classic and advanced control (e.g. *CartPole*, *LunarLander*), Atari environments (*MsPacman*, *Breakout*).
* You are missing a "." line 316.

**Strengths Contributions:**

The contribution is clear, gymnasium is probably the most used RL benchmark for RL researchers.
On the paper itself, apart from minor things, the paper is clear and well structured. It will allow non-expert RL users to easily understand the library.
The explanations on the functional design choices, as well as the difference between the necessity of separating truncation from termination are clear and convey the intuition on these choices, same for the part on vectorization.

---

> ### Author Rebuttal · Authors · 2025-07-31
>
> Thanks for the review, recognising the clear contributions we’ve made through the project and understanding the design decisions we’ve made.
>
> To answer your questions
> 1. We’ll add a link to the third-party project that explains how to add an environment to the list
> 2. We’ll look to upgrade the images to PDF or SVG
> 3. We’ll fix the minor grammatical errors
>
> On the additional feedback
> 1. Thanks for linking the post. We’ll add the range of projects that have been built off the original environments, in particular Atari and robotics-based environments.
> 2. Looking at the GPU emulated Atari project, it seems unmaintained for several years; however, there has been recent work in Arcade-Learning-Environments that implements the Atari environments to add jax jit support, which should allow significantly faster GPU integration.
> 3. Yes, we will keep `gymnasium[atari]` due to Atari’s historic importance to the project and for ease of use; however, the code and documentation for Atari are now completely separated out to their own repository and website. We took this decision as the original implementation basically hacked support into the library as a special case, and believed that a simpler library that was easier to maintain was better.
>
> We hope these answer your questions.

---

> > ### Comment · Area_Chair_wWuM · 2025-08-03
> >
> > Thanks for your rebuttal, RBDF do you have any comment?

---

### Official Review · Reviewer_XJcS · 2025-07-02

**Rating:** 6
**Confidence:** 4

**Summary:**

This paper details the design of a popular Reinforcement Learning training platform, Gymnasium, which extends prior OpenAI Gym platform with functionalities and usability.

**Dataset Code Accessibility:**

Yes

**Ethical Considerations:**

No, there are no or only very minor ethics concerns

**Final Justification:**

This paper details the design of a popular Reinforcement Learning training platform, Gymnasium, which extends prior OpenAI Gym platform with functionalities and usability. I am satisfied by the authors rebuttal and thus will keep my current score.

**Limitations Weaknesses:**

I only have a few suggestions on the development of the library in the future.
1. From my personal experiences, batched environments are very promising for training agents in a large scale. This is slightly different from the concept of vectorized environments where each environment is handled by a separate thread or process. While the support for vectorized environments are great in Gymnasium, I feel the support for batched environments are quite limited. Examples include IsaacLab, a popular robotics training backend. IsaacLab can simulate thousands of environments at the same time, on dedicated GPUs. But it is rather not easy to fit IsaacLab into either a normal environment, or a vector environment, because it indeed contains multiple environments, some of which needs resets from time to time without affecting others. Additionally, the termination (espeically with the terminal observations) can not be easily handled, which requires looping at the current moment, or wait for thirdparty training library to handle terminal observations differently. I think this is a missed opportunity.
2. Since training usually involves pytorch or Jax, I think adding more support for integration with either (or both) of these frameworks might increase the usability of the library. Often times, I find myself dealing with rountine tasks of "transferring this observation which is a dict of numpy array into dict of pytorch tensors and bring them onto the right device". This not only complicates the library development, also creates confusion during deployment, as the policy's output are usually pytorch tensor/jax and the training library will sometimes automatically translate them into numpy array.

**Strengths Contributions:**

Building upon the deprecated OpenAI Gym, Gymnasium has become the de facto RL training platform across numerous fields. The documentation of the library is clear and helpful. It is very user friendly with very mild learning curve. I personally find the extension on the API with termination and truncation very helpful and practical, considering the fact that we often deal with episodic MDPs.

Additionally, the work in Gymnasium with extensible space definition are highly modular and configurable, which is very welcomed in fields such as robotics where observation space can be highly heteogeneous, ranging from a simple list to nested dictionary of pytorch tensors.

Lastly, the vectorization function is Gymnasium provides an easy and standard way to construct and manage vectorized environments. Scalable environment contruction is crucial in training modern robotics controllers. For this I am happy to see the embrance for a modernized API.

---

> ### Author Rebuttal · Authors · 2025-07-31
>
> Thanks for the positive review and for agreeing with the improvements we’ve made to the project over the years.
>
> To answer your Limitations / Weaknesses
> * I believe with the recent changes to vector environments in v1.0 that the types of environments described in IsaacLab would be compatible with the API; however, we will reach out and see if this is compatible with the current implementation or what we can do to support it.
> * Thanks for raising this issue, as we similarly recognise it as a problem that can be complicated using the library, where almost all environments expect numpy arrays, however neural networks return Jax or Torch tensors. To support this, we recently added `TorchToNumpy` and `JaxToNumpy` wrappers, which should abstract away this complexity, particularly when using complex nested actions or observations, removing the need for users to implement this themselves. We’ll look to increase documentation of this to help users.
>
> We hope these answer your questions.

---

### Decision · Program_Chairs · 2025-09-18

**Decision:**

Accept (spotlight)

**Comment:**

This paper presents Gymnasium, the widely-adopted successor to OpenAI's deprecated Gym library and one of the current de facto standard APIs for RL environments. All reviewers recognized Gymnasium's fundamental importance. Authors adequately addressed all the issues that were raised, and will update the camera-ready accordingly.